# Risk Factors for Tooth Loss in Patients Undergoing Mid-Long-Term Maintenance: A Retrospective Study

**DOI:** 10.3390/ijerph17176258

**Published:** 2020-08-27

**Authors:** Hiroo Kawahara, Miho Inoue, Kazuo Okura, Masamitsu Oshima, Yoshizo Matsuka

**Affiliations:** 1Department of Stomatognathic Function and Occlusal Reconstruction, Graduate School of Biomedical Sciences, Tokushima University, 3-18-15 Tokushima, Tokushima 770-8504, Japan; hiro32@bronze.ocn.ne.jp (H.K.); inoue.miho@tokushima-u.ac.jp (M.I.); okura.kazuo@tokushima-u.ac.jp (K.O.); m-oshima@tokushima-u.ac.jp (M.O.); 2Kawahara Dental Clinic, 1-128 Muneshige, Mima, Tokushima 771-2104, Japan

**Keywords:** dental maintenance, patient age, remaining teeth, tooth loss

## Abstract

In this retrospective study, we identified risk factors for tooth loss in patients undergoing mid–long-term maintenance therapy. We surveyed 674 maintenance patients for ≥5 years after active treatment who visited a dental clinic between January 2015 and December 2016. Of these, 265 were men (mean age 54.6 ± 8.0 years old) and 409 were women (mean age 54.0 ± 7.9 years old). Study variables included patient compliance, sex, number of teeth lost, cause of tooth loss (dental caries, periodontal disease, root fracture, others, vital or non-vital teeth), age at start of maintenance, number of remaining teeth at start of maintenance, smoking, use of salivary secretion inhibitors, presence of diabetes mellitus, condition of periodontal bone loss, and use of a removable denture. Most lost teeth were non-vital teeth (91.7% of all cases) and the most common cause of tooth loss was tooth fracture (62.1% of all cases). A statistically significant risk factors for tooth loss was number of remaining teeth at the start of maintenance (*p* = 0.003).

## 1. Introduction

Several dental studies have shown that the most common cause of tooth loss in adults is dental caries, followed by periodontal disease [1,2,3,4,5]. Thus, the control of dental caries and periodontal disease is critical for the prevention of tooth loss. A major etiological factor underlying the pathogenesis of dental caries and periodontal disease is the formation of a biofilm on the tooth surface. Regular maintenance with biofilm removal is critical for preventing dental caries and periodontal disease. Axelsson and Lindhe [6] and Axelsson et al. [7] found that regular maintenance therapy reduced the incidence of dental caries and periodontal disease, significantly lowering the risk of tooth loss. Other studies have also shown decreased tooth loss with regular maintenance therapy [8,9,10,11,12,13]. Axelsson et al. [7] reported that the most frequent cause of tooth loss in patients undergoing maintenance therapy was root fracture. Other studies showed that the most frequent cause of tooth loss, even with maintenance therapy, was periodontal disease [14,15]. Various factors, such as age, smoking, the presence of diabetes mellitus (DM), and others, are also related to tooth loss, and the risk of tooth loss increases with the presence of additional non-vital teeth (e.g., those that have undergone root canal treatment) [14,16,17,18]. Despite reports that various factors affect tooth loss, many of these studies were conducted under special circumstances, such as at a university hospital or within a specialist clinic, where patient demographics were limited. In Japan, most dental patients visit the private dental clinics, and physically or mentally challenged patients go to specialist clinics. Additionally, few studies on the risk factors for tooth loss have used large sample sizes. Furthermore, differences in patients’ immunological and genetic backgrounds, cultural factors, and socioeconomic features may affect the risk of tooth loss [5,19,20].

There have been several large-sample-size reports of tooth loss under long-term maintenance in Japan. One study compared the number of lost teeth among different levels of compliance with maintenance [21]; another study investigated whether the number of non-vital teeth was an indicator of lost teeth during maintenance [18]. A questionnaire survey was used to investigate the risk factors for tooth loss, but tooth loss under maintenance was not examined [22]. Few studies have used a large sample size when investigating risk factors for tooth loss under maintenance. These studies were mostly conducted at universities, not private dental clinics.

Therefore, we conducted a retrospective analysis of patients at a private Japanese general clinic to verify the results of these previous surveys and investigate risk factors for tooth loss under mid–long-term maintenance.

## 2. Materials and Methods 

### 2.1. Study Design and Patient Sampling

This retrospective study was conducted with data from 674 healthy patients selected from the patient records of one general dentist (Hiroo Kawahara) in a Japanese private dental clinic.

Patients visited the clinic for dental maintenance from January 2015 to December 2016. Their age at baseline was 40‒69. An explanation of the baseline is given in Section 2.3 below. The observation period was the number of years from the baseline to the end of 2016. At the end of 2016, patients were undergoing dental maintenance for ≥5 years after establishment of their baseline. The observation period and the number of maintenance years were the same. The number of teeth lost from baseline to the end of 2016 was measured (Figure 1). Dental examinations were performed and associated records were obtained by the dentist (Hiroo Kawahara), with the assistance of several trained dental hygienists. Patients who received implant treatment and those with no teeth were excluded because the number of remaining teeth were surveyed as a study variable.

### 2.2. Dental Treatment up to Maintenance 

At the first visit, all patients underwent a general health and dental interview. Patients then received a full-jaw X-ray, photographs, and periodontal examination. After a treatment plan was developed by the dentist (Hiroo Kawahara) and the patient’s consent was obtained, active treatment was performed. With active treatment, all patients initially received education about dental caries, periodontal disease, and other oral diseases. They underwent individual risk assessment to learn about risk control for dental caries and periodontal disease. Cariogram computer software (D Bratthall, Malmo, Sweden) [23] was used for caries risk assessment, and the Oral Health Information Suite (PreViser Corp., Concord, NH, USA) [24] was used for periodontal disease risk assessment. Following patient education, active treatments such as tooth extraction, restorative, endodontic, and periodontal therapy were performed. The periodontal therapy involved oral hygiene enhancement, scaling and root planing, and/or surgical therapy according to the individual patient’s needs.

### 2.3. Baseline Status

Following active treatment, patients underwent reevaluation of clinical parameters including probing depth, bleeding on probing, plaque index, and number of teeth present. The reassessment of clinical parameters was used as the baseline for each patient. The duration of maintenance was measured from the baseline (Figure 1).

### 2.4. Maintenance Criteria, Procedure, Interval, and Compliance

All patients were required to meet the following five criteria established by Miyamoto et al. [25] at baseline: (1) <10% of sites with bleeding on probing; (2) an overall plaque score of <15%; (3) <10% of sites with a probing depth of ≥4 mm; (4) no defective restorations; and (5) no active dental caries. Patients who failed to meet the criteria were treated again with active treatment.

The condition of periodontal bone loss in patients was divided into four classes. First, using a dental X-ray photograph, the bone absorption degree of each tooth was judged according to the following criteria: 0, no bone absorption; 1, bone absorption less than 1/3 of the root; 2, bone absorption is 1/3 or more and less than 1/2 of the root; and 3, bone absorption is 1/2 or more of the root. Then, the total score was divided by the number of teeth to determine the following: Class 0 = 0; Class 1 = 0 or more and less than 1; Class 2 = 1 or more and less than 2; Class 3 = 2 or more.

After reevaluation, all patients who met the criteria underwent maintenance, which consisted of a full-mouth clinical examination, supragingival scaling and polishing, and subgingival debridement using Gracey curettes, an ultrasonic scaler, rubber cups, and abrasive paste. Oral hygiene instructions and application of fluoride were provided according to the patient’s individual needs. When a need for treatment was determined during maintenance, the dentist (Hiroo Kawaharra) provided the appropriate treatment (tooth extraction, restoration, prosthetic rehabilitation, and/or endodontic treatment).

Maintenance intervals were based on the patient’s periodontal status at baseline. Patients with periodontal pockets ≥4 mm at baseline were assigned to maintenance therapy at 3-month intervals. Patients without pockets ≥4 mm were assigned maintenance therapy at 6-month intervals. There were no changes to the maintenance intervals during maintenance.

Patient compliance during maintenance was divided into the following two categories, according to the classification established by Miyamoto et al. [25]. Regular maintenance indicated that the following two requirements were met: (1) attending ≥70% of expected maintenance visits; and (2) the interval between visits during maintenance did not exceed a maximum of 2 years. Irregular maintenance indicated that the criteria for regular compliance were not met but continued maintenance visits were carried out.

### 2.5. Cause and Status of Tooth Loss under Maintenance

The cause of any tooth loss was determined by the dentist (Hiroo Kawahara) during extraction. These causes included dental caries, periodontal disease, root fracture, and others (e.g., apical lesion, trauma, tooth transposition, or tooth extraction for convenience). These statuses included vital teeth (teeth with living dental pulp) and non-vital teeth (teeth with non-vital dental pulp; e.g., those that have undergone root canal treatment). The extraction of wisdom and deciduous teeth was excluded from the analysis. We excluded tooth extraction after baseline that was planned to be extracted before baseline by the initial treatment plan. This was because the patient’s consent was not obtained before baseline. 

### 2.6. Data Collection

Data of the following 14 characteristics were collected: compliance, sex, age at baseline, number of remaining teeth at baseline, number of teeth lost to various causes (dental caries, periodontal disease, root fracture, others) from baseline to the end of 2016, status of lost teeth (vital/non-vital), condition of periodontal bone loss at baseline (class 0, 1, 2, 3), use of removable dentures at baseline (yes/no), smoking (yes/no), use of salivary secretion inhibitors (SSIs) (e.g., antidepressants, anxiolytics, diuretics, antihypertensives, antiarrhythmic drugs, and other drugs that inhibit salivary secretion) (yes/no), and presence of DM (yes/no). Patients with DM were well controlled by their physicians (hemoglobin A1c < 7%, National Glycohemoglobin Standardization Program). Smoking, use of SSIs, and presence of DM were recorded at baseline. 

### 2.7. Statistical Analysis

Logistic regression analysis was performed for the relationship between the presence or absence of tooth loss and the following factors: compliance (regular/irregular), sex (male/female), smoking (yes/no), use of SSI (yes/no), presence of DM (yes/no), age at baseline (40‒54/55–60 years), remaining teeth at baseline (28–25/≤24 teeth), use of removable dentures at baseline (yes/no) and condition of periodontal bone loss at baseline (Class 0–1/Class 2‒3). First, we performed a logistic regression analysis for each variable, then we performed a multiple logistic regression analysis. The variables at baseline were selected to define risk factors for tooth loss under maintenance. Therefore, the cause and status (vital or non-vital) of tooth loss were not selected as variables.

All statistical analyses were performed with JMP version 14 (SAS Institute, Cary, NC, USA), and *p* < 0.05 indicated statistical significance. The statistical analysis was not verified by an independent statistician.

### 2.8. Ethics Approval and Consent to Participate

This study was approved by the Clinical Research Ethics Review Committee of Tokushima University Hospital (approval number: 2674). In this study, the patients’ right to privacy protection was respected; additionally, written informed consent was obtained from all patients. This research was conducted in full accordance with the Declaration of Helsinki established by the World Medical Association.

## 3. Results

Characteristics of the sample data in this study are shown in Table 1. The study sampled 674 patients and 572 lost teeth over the observation period. The number of years from baseline was the observation period.

The distribution of the observation period is shown in Table 2. The mean observation period was 9.3 years. The observation period was from baseline to the end of 2016.

Tooth loss according to cause and vital versus non-vital tooth status are shown in Table 3. Most lost teeth were non-vital teeth (91.7% of all cases); the most common cause of tooth loss was tooth fracture (62.1% of all cases). 

Characteristic study variables and relationships with tooth loss are shown in Table 4. The variables at baseline were selected to define risk factors for tooth loss under maintenance. Therefore, the cause and status (vital or non-vital) of tooth loss were not selected as variables. Logistic regression analysis was performed for each variable. There were significant differences between the following variables and the presence or absence of tooth loss: use of SSI, age at baseline, remaining teeth at baseline, and use of removable dentures at baseline.

The results of multiple logistic regression analyses of study variables with tooth loss showed that number of remaining teeth at baseline was significantly associated with the presence or absence of tooth loss (Table 5). 

## 4. Discussion

We performed this retrospective study to investigate tooth loss in 674 patients undergoing mid–long-term maintenance therapy at a private dental clinic in Japan. 

Root fractures were the most common cause of tooth loss, followed by caries and periodontal disease. These results are similar to those from a study in a general dental clinic reported by Axelsson et al. [7]. In a study showing that the primary cause of tooth loss was periodontal disease, data samples were obtained from the private offices of three periodontists [14]. In another study, patient data were obtained after treatment by a periodontist [15]. In these studies, most patients were considered high risk for periodontal disease. In this study, 39.2% of patients were included in classes two and three in the periodontal bone loss status at baseline. Few patients had severe periodontal bone loss. There was no significant difference in patient periodontal bone loss status and the presence or absence of tooth loss. Some patients might have been high risk for periodontal disease. Therefore, periodontal disease might not have been the common cause of tooth loss.

In our study, 91.7% of tooth loss involved non-vital teeth. This may be explained by a previous study in which root canal treatment had a significant effect on tooth loss [17]. Root canal treatment might also affect the deterioration of furcation lesions in molars [17]. One study showed that as the number of non-vital teeth increased, tooth loss caused by root fractures and caries increased [18]. Therefore, a decrease in the number of non-vital teeth may reduce tooth loss caused by caries, periodontal disease, and root fractures. Decreasing the number of non-vital teeth may reduce tooth loss in maintenance. 

There were no significant differences between patient compliance and the presence or absence of tooth loss in this study. A study investigating the impact of compliance on tooth loss during maintenance found that the higher the compliance, the lower the risk of tooth loss [26]. It was suggested that the effect of compliance in preventing tooth loss may result from the effectiveness of maintenance therapy as well as stabilization of self-discipline in maintaining oral hygiene [26]. Therefore, the following factors might explain this result. (1) The levels of oral hygiene in both compliance groups were similar. (2) Patients were under continuous maintenance, if not completely compliant; however, there was more tooth loss in the irregular compliance group, although this did not reach statistical significance.

There was no significant sex difference related to the presence or absence of tooth loss. Several studies reported that men lose more teeth than women [13,21,27] for the following reasons: (1) men are less likely to adhere to maintenance programs than women; and (2) risk behavior factors such as smoking are more common in men than women. In this study, women had less tooth loss than men, although this difference was not statistically significant. Although there were more women than men in this study, the two reasons stated above might explain this finding.

We found no significant differences in the effect of smoking on tooth loss under maintenance. Smoking is, however, a clear risk factor for periodontal disease [28]. In a previous study, smokers exhibited significantly more tooth loss than nonsmokers [13]. The presence or absence of DM was not significantly different for tooth loss. DM is also a risk factor for periodontal disease [28]. Previous studies reported that patients with DM lost significantly more teeth than those without DM [13,29]. In this study, tooth loss due to periodontal disease was only 11.5% of the total. Most causes of tooth loss were due to caries and root fractures. This study may have had a small number of patients at high-risk of periodontal disease. That may be the reason why smoking and DM did not affect tooth loss. It was also helpful that all patients with DM were in good condition.

Previous studies have also shown significant associations between tooth loss and systemic diseases such as hypertension, heart disease, cerebrovascular disease, rheumatoid arthritis, and asthma [4,14]. Although the relationship between the etiology of disease and tooth loss is clear with regard to DM, this relationship remains unclear for many other diseases. The adverse effect of thirst was reportedly associated with the therapeutic drugs used for many diseases [30]. However, no reports have described the relationship between SSI use and tooth loss. The present study showed a significant association between SSI use and the presence or absence of tooth loss.

The number of remaining teeth at baseline was significantly different between groups with 28–25 or ≤ 24 teeth remaining. It was previously reported that tooth loss was increased by reducing the number of remaining teeth [21,31,32]. According to multiple logistic regression analysis in this study, only the number of remaining teeth at baseline was significantly associated with the presence or absence of tooth loss in accord with a previous study [32]. One explanation for the increase in tooth loss as the number of remaining teeth decreases may be the use of fixed or removable dentures. We excluded patients treated with implants from this study. The use of dentures increased as the number of remaining teeth decreased. Several studies reported a significant loss of denture abutments [33,34]. In this study, patient use of removable dentures was significantly associated with the presence or absence of tooth loss. Another explanation may be that patients with a low number of remaining teeth had received more restorative prosthetic treatment and therefore had more non-vital teeth. Some reports have also described the significant loss of non-vital teeth [17]. 

In this study, as in previous reports, there was more tooth loss in the elderly group [7,11,12]. In the logistic regression analysis, being elderly was significantly associated with the presence or absence of tooth loss. However, multiple logistic regression analysis indicated no association between age and the presence or absence of tooth loss. Other studies have also reported that tooth loss and age may not be related [32]. Increased tooth loss in the older group may be due to a decrease in the number of remaining teeth. 

This study had some limitations. First, the study was conducted in the private practice of a general dentist, so the periodontal status of all patients was healthy-to-moderate for most of the patient population. Few patients had a severe periodontal condition. Periodontal status might explain the differences from previous reports. Second, our results can be explained by other unmeasured risk factors for tooth loss (e.g., education level). Previous studies reported that lower education levels were associated with tooth loss [35]. Third, our data did not include salivary output. It was reported that patients with Sjogren’s syndrome, where salivary secretion is low and the main symptom is dry mouth, have significantly greater tooth loss than those without Sjogren’s syndrome [36]. If saliva production was measured, the association with tooth loss might have been clearer. Another limitation of this study is that despite its large sample size, the sample was from a single clinic in Japan. Nevertheless, this type of study including patient data with a large sample size from one clinic might provide important information for public health, health systems, and epidemiological research.

## 5. Conclusions

We found that most lost teeth were non-vital teeth and that tooth fracture was the most common cause of tooth loss under maintenance. We also found that a statistically significant risk factor for tooth loss was the number of remaining teeth at the start of maintenance.

## Figures and Tables

**Figure 1 ijerph-17-06258-f001:**
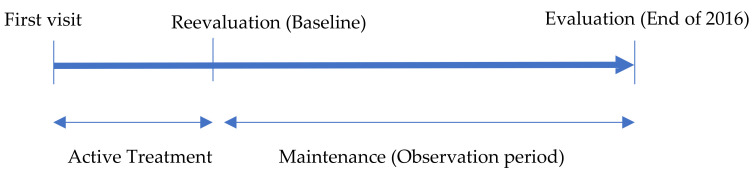
Time table of active treatment and maintenance.

**Table 1 ijerph-17-06258-t001:** Characteristics of patients in this study.

Sample Size	Age (Years)	Year (Years)	RT (Number of Teeth)	Total Number of Teeth Lost	Cause of Tooth Loss				Tooth Loss/Year Per Patient
					Dental Caries	Periodontal Disease	Root Fracture	Other	
N	Mean ± SD			Number of Teeth					
674	54.3 ± 8.0	9.3 ± 2.6	23.0 ± 5.5	572	127	66	355	24	0.09

Abbreviations: SD, standard deviation; Age, age at baseline; RT, remaining teeth at baseline; Year, years from baseline.

**Table 2 ijerph-17-06258-t002:** Distribution of the observation period.

Observation Period (Years)	5	6	7	8	9	10	11	12	13	14	Total
Number of Patients	75	84	63	102	112	68	43	46	42	39	674

**Table 3 ijerph-17-06258-t003:** Tooth loss according to cause and vital versus non-vital tooth status.

Cause of Tooth Loss	Number of Vital Teeth (%)	Number of Non-Vital Teeth (%)	Total Number of Teeth (%)
Dental caries	2 (0.3%)	125 (21.9%)	127 (22.2%)
Periodontal disease	31 (5.4%)	35 (6.1%)	66 (11.5%)
Root fracture	3 (0.5%)	352 (61.5%)	355 (62.1%)
Other	11 (2.0%)	13 (2.3%)	24 (4.2%)
Total (%)	47 (8.2%)	525 (91.7%)	572 (100%)

**Table 4 ijerph-17-06258-t004:** Characteristics of study variables and relationships with tooth loss.

Variables		Sample Size	Age (Years)	Year (Years)	RT (Number of Teeth)	Tooth Loss (Number of Teeth)	Tooth Loss /Patient/Year (Number of Teeth)	Logistic Regression Analyses	
		*N*	Mean ± SD	Mean ± SD	Mean ± SD			Odds Ratio (95% CI)	*p*-Value
Compliance	Regular	636 (94.4%)	54.2 ± 7.9	9.3 ± 2.5	23.0 ± 5.4	528 (92.3%)	0.09	1	
	Irregular	38 (5.6%)	54.7 ± 7.6	7.8 ± 2.0	22.2 ± 6.0	44 (7.7%)	0.15	1.89 (0.97–3.66)	0.06
Sex	Male	265 (39.3%)	54.6 ± 8.0	9.1 ± 2.6	22.8 ± 5.4	238 (41.6%)	0.10	1	
	Female	409 (60.7%)	54.0 ± 7.9	9.3 ± 2.5	23.0 ± 5.4	334 (58.4%)	0.09	0.82 (0.60–1.12)	0.20
SM	No	548 (75.5%)	54.6 ± 7.8	9.2 ± 2.5	23.0 ± 5.5	478 (83.6%)	0.09	1	
	Yes	126 (18.7%)	52.6 ± 8.1	8.7 ± 2.5	22.4 ± 5.5	94 (16.4%)	0.09	0.92 (0.62–1.36)	0.66
SSI	No	509 (75.5%)	52.9 ± 7.8	9.0 ± 2.5	23.2 ± 5.4	503 (88.0%)	0.09	1	
	Yes	165 (24.5%)	58.2 ± 6.8	9.9 ± 2.5	22.2 ± 5.3	69 (12.0%)	0.16	1.53 (1.07–2.17)	0.02
DM	No	628 (93.2%)	53.9 ± 8.0	9.2 ± 2.5	23.0 ± 5.4	503 (88.0%)	0.09	1	
	Yes	46 (6.8%)	58.2 ± 6.0	9.6 ± 2.6	21.5 ± 5.3	69 (12.0%)	0.16	1.79 (0.98–3.28)	0.06
AGE	40–54	332 (49.3%)	47.4 ± 4.4	9.2 ± 2.6	24.7 ± 4.2	197 (34.4%)	0.06	1	
	55–69	342 (50.7%)	61.0 ± 3.9	9.3 ± 2.6	21.3 ± 5.9	375 (65.6%)	0.18	1.58 (1.17–2.15)	0.003
RT	28–25	366 (54.3%)	51.7 ± 7.9	9.2 ± 2.6	26.7 ± 1.1	195 (34.1%)	0.06	1	
	≤24	308 (45.7%)	57.4 ± 6.8	9.4 ± 2.6	18.5 ± 5.2	377 (65.9%)	0.13	2.33 (1.71–3.20)	≤0.001
BL	Class 0–1	410 (60.8%)	52.7 ± 8.2	9.2 ± 2.6	24.3 ± 4.6	315 (55.1%)	0.08	1	
	Class 2–3	264 (39.2%)	56.7 ± 7.0	9.4 ± 2.6	20.9 ± 6.0	257 (44.9%)	0.10	1.17 (0.86–1.61)	0.31
RD	No	486 (69.4%)	52.5 ± 8.0	9.2 ± 2.6	26.0 ± 1.7	290 (50.7%)	0.07	1	
	Yes	206 (30.6%)	58.2 ± 6.3	9.4 ± 2.6	16.1 ± 4.8	283 (49.3%)	0.15	2.18 (1.56–3.03)	≤0.001

Abbreviations: SD, standard deviation; Age, age at baseline; RT, remaining teeth at baseline; Year, years from baseline; SM, smoking; SSI, use of salivary secretion inhibitors; DM, diabetes mellitus; BL, condition of periodontal bone loss at baseline; BD, use of removable denture at baseline; CI, confidence interval.

**Table 5 ijerph-17-06258-t005:** Multiple logistic regression analyses of study variables with tooth loss.

Variables		Coefficient	Standard Error	χ^2^	Odds Ratio (95% CI)	*p*-Value
Compliance	Regular				1	0.09
	Irregular	0.29	0.17	2.85	1.80 (0.91–3.57)	
Sex	Men				1	0.13
	Women	−0.14	0.09	2.25	0.76 (0.52–1.09)	
SM	No				1	0.23
	Yes	−0.14	0.12	1.42	0.75 (0.48–1.20)	
SSI	No				1	0.23
	Yes	0.12	0.09	1.43	1.26 (0.86–1.85)	
DM	No				1	0.40
	Yes	0.14	0.16	0.71	1.32 (0.69–2.52)	
AGE	40–54				1	
	55–69	0.12	0.18	0.15	1.13 (0.80–1.60)	0.47
RT	28–25				1	
	≤24	0.68	0.23	8.73	1.99 (1.26–3.14)	0.003
BL	Class 0–1				1	
	Class 2–3	0.15	0.17	0.80	0.85 (0.60–1.21)	0.37
RD	No				1	
	Yes	0.11	0.12	0.79	1.25 (0.76–2.03)	0.37

Abbreviations: SD, standard deviation; AGE, age at baseline; RT, remaining teeth at baseline; SM, smoking; SSI, use of salivary secretion inhibitors; DM, diabetes mellitus; BL, condition of periodontal bone loss at baseline; RD, use of removable denture at baseline; CI, confidence interval.

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
