# Peer review of "Risk Factors for Tooth Loss in Patients Undergoing Mid-Long-Term Maintenance: A Retrospective Study"

_ijerph, 2020, doi:10.3390/ijerph17176258_

Round 1

Reviewer 1 Report

GENERAL COMMENTS

The present study investigated risk factors associated with tooth loss in patients undergoing long-term maintenance. The authors found that several factors were associated with tooth loss.

I think this study is interesting. Although the topic may be considered of interest, several issues raised in this manuscript (see comments below).

Major concerns

  1. Before undergoing maintenance, it is reasonable to assume that teeth with high risk for loosing would be extracted. Therefore, the conditions of teeth (for example, the condition of bone loss) before undergoing maintenance would be crucial for this kind of studies. However, in this study, there are few information for these matters were not mentioned. I recommend the authors to add these variables for analysis.

  1. This study was performed in one single dental practice in Japan. Therefore, selection bias could affect the result of this study. I wonder the situation of tooth loss in Tokushima prefecture compared with other prefectures in Japan.

  1. I could not understand why the authors mentioned that “Starting maintenance therapy earlier may prevent future tooth loss.” In the abstract. This study was retrospective and did not perform any comparison of the timing of the initiation of maintenance.

  1. In this study, I assume that there are 3 groups; regular maintenance group, irregular maintenance group and non-maintenance group. However, only regular maintenance group was mainly analyzed in this study. The authors should clarify these groups and if they set these groups, they should compare the differences among these groups.

  1. I recommend the authors to check the STROBE statement.

Abstract

  1. P1 L19: Please unify the expression of age (years or years old).
  2. As study variables, the information of vital or non-vital was not included in P1 L16-18. However, non-vital teeth is written in P1 L22.
  3. I could not understand why the result of this study could lead to “Starting maintenance therapy earlier may prevent future tooth loss”.

Introduction

  1. P1 L43: The authors mentioned that “such as at a university hospital or within a specialist clinic, where patient demographics were limited.” It is reasonable for the readers to assume that patient demographics would not be limited in this study. However, this study was conducted in one single private dental clinic. I think the authors should show why patient demographics were not limited in this study.

  1. I would like to know why the authors mentioned about the National Health Insurance System in Japan. Did the authors think that this system would be associated with tooth loss? Behavior of the patients would be differed compared with other countries?

  1. P2 L60: Is that possible “to determine the most convenient and effective methods for preventing tooth loss” by using this study design?

Materials and Methods

  1. L65: In Japan, almost all private dental clinics would be general dental clinics. Did the authors had specific reasons for selecting Kawahara Dental Clinic?
  2. I recommend the authors to add a flowchart for the criteria of the participants.
  3. L71 and L75: As for non-maintenance patients, 106 or 104?
  4. L78: The authors should explain why they excluded patients with implant treatment.
  5. L82: The dentist means H.K.?
  6. L121: I could not understand this sentence. The patient’s consent would always be obtained when dentists extract their teeth.
  7. L129: As for diabetes mellitus, did the authors obtain the laboratory values such as HbA1c? Only self-report information would not be enough to determine whether the patients had DM or not.
  8. Why the variables between MP5 and NMP5 were different?
  9. The authors should explain why they chose nonparametric statistics.
  10. L150: The authors should clarify the method of logistic regression analysis (I assume the forced entry method was used).
  11. As stated above, the readers might confuse why there were 3 groups but only MP5 were analyzed in this study. I strongly recommend the authors to reconsider the statistical approach.
  12. The information of vital or non-vital teeth should be explained. How the authors decided this matter?

Results

  1. I recommend to put percentage to all Tables especially the cause of tooth loss.
  2. Table2, 3, 4 should be explained in the result section.
  3. L172-175: There were no explanation for these information in the method section. How the authors obtained these information? Were these information intended to be variables for the analysis?
  4. I could not understand Table5. The authors should clarify where the statistical differences were observed in the Table.
  5. As for Table6, the authors should explain why they chose these variables as independent variables. The other information such as the number of non-vital teeth might be associated with tooth loss?
  6. L198-200: I could not understand this sentence, if there was correlation between factors, these factors should not be included in the logistic regression due to multicollinearity. The authors mentioned that they did not include 2 variables because there was no correlation.
  7. L198-203: These information should not be explained in the result section.
  8. L207: 95% CI should be mentioned.

Discussion

  1. As mentioned above, the author should clarify which groups they mentioned in the discussion (regular maintenance group, irregular maintenance group and non-maintenance group). Moreover, the authors should show appropriate data for the discussion.
  2. L216-230: The authors compared between MP5 and NMP5. However, there are few data as for NMP5. If the authors has intention to compare the difference between MP5 and NMP5, they should show appropriate data (for example, there might be more DM patients among NMP5 compared with MP5.).
  3. L235-236: I could not find this information in the method or result section. All data mentioned in the discussion section should be shown in the result section.
  4. L254: The authors had the information of oral hygiene? If so, this should be added to the variables.
  5. L262: Although the authors mentioned that smoking might be associated with tooth loss, the result showed there was no association between tooth loss and smoking (Table6). Does the authors have any reasons?
  6. L267-269: I could not find the average age of smokers in the result section. Moreover, the authors should explain the average age affected the result of this study more in details.
  7. L270-271: This is a scientific article. Therefore, I think this sentence should not be in placed because this information is nothing do with the result of this study. If this information was associated with the result of this study, the authors should explain.
  8. L272-274: The authors should discuss more about DM. For example, why patients with DM had more tooth loss even though they receive maintenance?
  9. L281-283: How the authors mentioned this sentence? This study just showed the association between tooth loss and SSI. I do not think the authors could mention “significant 281 relationship between various systemic diseases and tooth loss is not a result of the disease etiology” from the results of this study.
  10. L289-297: These sentences should be moved to the result section.
  11. L293-294: I could not find the result of this statistical analysis.
  12. L298-300: I could not understand this sentence.
  13. L301-302: I do not agree with the authors for this sentence. This cutline was set by the authors. If the cutline was changed, this result could be also changed.
  14. L305: If the authors had the information of the use of dentures, why don’t you include it to the analysis?
  15. L310: There are several articles reporting the relationship between the number of remaining teeth and tooth loss. (https://pubmed.ncbi.nlm.nih.gov/25765572/)
  16. L315-321: These sentences should be moved to the result section.
  17. L324-325; I could not understand this sentence.
  18. I think there are more limitations in this study.

Reviewer 2 Report

Comments to authors

There is a great need or research investigating the effect of long-term maintenance, so this report (“Risk factors for tooth loss in patients undergoing long-term maintenance: A retrospective study”) will be of interest to readers of IJERPH. However, there are several issues that need to be addressed before this report can be accepted.

1. There were no data about the observation period of each participant. Please include data on the distribution of observation periods.

2. How did you count the number of teeth lost? Did you count the number of teeth lost during the two years from January 2015 to December 2016? If not, you should report the observation period for the tooth loss data. Please provide a detailed explanation of how you calculated teeth lost per person per year.

3. Your study title refers to “long-term maintenance”, but a maintenance period of 5 years is not long-term compared with previous reports.

4. It is inappropriate to include the data of participants in their 20s and those in their 90s when calculating the tooth loss results. Please exclude the data of those under 40 and the data of the extremely elderly participants. Previous reports have established that tooth loss is affected by number of remaining teeth.

Abstract

5. The title of this study was “Risk factors for tooth loss in patients undergoing long-term maintenance: A retrospective study”. However, risk factors are not addressed in the Abstract.

Methods

6. Why did you classify the participants into 3 age groups in Table 6, but 5 groups in Table 3?

7. It is difficult to identify the risk factors of tooth loss with only logistic regression analyses. Please use multiple logistic regression analysis.

Results

8. The order of the Tables is wrong.

9. There is no explanation of the results displayed in Tables 1 through 4. Please add explanations of the results shown in those Tables.

10. I think Tables 4 and 5 are unnecessary.

References

11. There are too many references. Please select only the necessary ones.

Round 2

Reviewer 1 Report

Unfortunately, my concerns were not revised appropriately.

If  the number of teeth at baseline was only associated with tooth loss during maintenance, you would not be able to modify this factor. 

In addition, there are several studies pointing out the relationship between the number of teeth at baseline and tooth loss.

Therefore, originality of this study would be relatively low for publication.

Reviewer 2 Report

The report was well revised with reviewer's comments.